# NEURAL LYAPUNOV MODEL PREDICTIVE CONTROL

## ABSTRACT

With a growing interest in data-driven control techniques, Model Predictive Control (MPC) provides a significant opportunity to exploit the surplus of data reliably, particularly while taking safety and stability into account. In this paper, we aim to infer the terminal cost of an MPC controller from transitions generated by an initial *unknown* demonstrator. We propose an algorithm to alternatively learn the terminal cost and update the MPC parameters according to a stability metric. We design the terminal cost as a Lyapunov function neural network and theoretically show that, under limited approximation error, our proposed approach guarantees that the size of the stability region (region of attraction) is greater than or equal to the one from the initial demonstrator. We also present theorems that characterize the stability and performance of the learned MPC in the presence of model uncertainties and sub-optimality due to function approximation. Empirically, we demonstrate the efficacy of the proposed algorithm on non-linear continuous control tasks with soft constraints. Our results show that the proposed approach can improve upon the initial demonstrator also in practice and achieve better task performance than other learning-based baselines.

## 1 INTRODUCTION

Control systems comprise of safety requirements that need to be considered during the controller design process. In most applications, these are in the form of state/input constraints and convergence to an equilibrium point, a specific set or a trajectory. Typically, a control strategy that violates these specifications can lead to *unsafe* behavior. While learning-based methods are promising for solving challenging non-linear control problems, the lack of interpretability and provable safety guarantees impede their use in practical control settings (Amodei et al., 2016). Model-based reinforcement learning (RL) with planning uses a surrogate model to minimize the sum of future costs plus a learned value function terminal cost (Moerland et al., 2020; Lowrey et al., 2018). Approximated value functions, however, do not offer safety guarantees. By contrast, control theory focuses on these guarantees but it is limited by its assumptions. Thus, there is a gap between theory and practice.

A feedback controller stabilizes a system if a local Control Lyapunov Function (CLF) function exists for the pair. This requires that the closed-loop response from any initial state results in a smaller value of the CLF at the next state. The existence of such a function is a necessary and sufficient condition for showing stability and convergence (Khalil, 2014). However, finding an appropriate Lyapunov function is often cumbersome and can be conservative. By exploiting the expressiveness of neural networks (NNs), Lyapunov NNs have been demonstrated as a general tool to produce stability (safety) certificates (Bobiti, 2017; Bobiti & Lazar, 2016) and also improve an existing controller (Berkenkamp et al., 2017; Gallieri et al., 2019; Chang et al., 2019). In most of these settings, the controller is parameterized through a NN as well. The flexibility provided by this choice comes at the cost of increased sample complexity, which is often expensive in real-world *safety-critical* systems. In this work, we aim to overcome this limitation by leveraging an initial set of one-step transitions from an *unknown* expert demonstrator (which may be sub-optimal) and by using a learned Lyapunov function and surrogate model within an Model Predictive Control (MPC) formulation.

Our key contribution is an algorithmic framework, *Neural Lyapunov MPC* (NLMPC), that obtains a single-step horizon MPC for Lyapunov-based control of non-linear deterministic systems with constraints. By treating the learned Lyapunov NN as an estimate of the value function, we provide theoretical results for the performance of the MPC with an imperfect forward model. These results complement the ones by Lowrey et al. (2018), which only considers the case of a perfect dynamics

model. In our proposed framework, alternate learning is used to train the Lyapunov NN in a supervised manner and to tune the parameters of the MPC. The learned Lyapunov NN is used as the MPC's terminal cost for obtaining closed-loop stability and robustness margins to model errors. For the resulting controller, we show that the size of the stable region can be larger than that from an MPC demonstrator with a longer prediction horizon. To empirically illustrate the efficacy of our approach, we consider constrained non-linear continuous control tasks: torque-limited inverted pendulum and non-holonomic vehicle kinematics. We show that NLMPC can transfer between using an inaccurate surrogate and a nominal forward model, and outperform several baselines in terms of stability.

## 2 PRELIMINARIES AND ASSUMPTIONS

**Controlled Dynamical System**    Consider a discrete-time, time-invariant, deterministic system:

$$x(t+1) = f(x(t), u(t)), \quad y(t) = x(t), \quad f(0,0) = 0, \tag{1}$$

where $t \in \mathbb{N}$ is the timestep index, $x(t) \in \mathbb{R}^{n_x}$, $u(t) \in \mathbb{R}^{n_u}$ and $y(t) \in \mathbb{R}^{n_y}$ are, respectively, the state, control input, and measurement at timestep $t$. We assume that the states and measurements are equivalent and the origin is the equilibrium point. Further, the system (1) is subjected to closed and bounded, convex constraints over the state and input spaces:

$$x(t) \in \mathbb{X} \subseteq \mathbb{R}^{n_x}, \quad u(t) \in \mathbb{U} \subset \mathbb{R}^{n_u}, \quad \forall t > 0. \tag{2}$$

The system is to be controlled by a feedback policy, $K : \mathbb{R}^{n_x} \to \mathbb{R}^{n_u}$. The policy $K$ is considered *safe* if there exists an invariant set, $\mathbb{X}_s \subseteq \mathbb{X}$, for the closed-loop dynamics, inside the constraints. The set $\mathbb{X}_s$ is also referred to as the *safe-set* under $K$. Namely, every trajectory for the closed-loop system that starts at some $x \in \mathbb{X}_s$ remains inside this set. If $x$ asymptotically reaches the target , $\bar{x}_T \in \mathbb{X}_s$, then $\mathbb{X}_s$ is a Region of Attraction (ROA). In practice, convergence often occurs to a small set, $\mathbb{X}_T$.

**Lyapunov Conditions and Safety**    We formally assess the safety of the closed-loop system in terms of the existence of the positively invariant-set, $\mathbb{X}_s$, inside the state constraints. This is done by means of a learned CLF, $V(x)$, given data generated under a (initially unknown) policy, $K(x)$.

The candidate CLF needs to satisfy certain properties. First, it needs to be upper and lower bounded by strictly increasing, unbounded, positive ($\mathcal{K}_\infty$) functions (Khalil, 2014). We focus on optimal control with a quadratic stage cost and assume the origin as the target state:

$$\ell(x, u) = x^T Q x + u^T R u, \quad Q \succeq 0, \ R \succ 0. \tag{3}$$

For above, a possible choice for $\mathcal{K}_\infty$-function is the scaled sum-of-squares of the states:

$$l_\ell \|x\|_2^2 \leq V(x) \leq L_V \|x\|_2^2, \tag{4}$$

where $l_\ell$ and $L_V$ are the minimum eigenvalue of $Q$ and a Lipschitz constant for $V$ respectively.

Further for safety, the convergence to a set, $\mathbb{X}_T \subset \mathbb{X}_s$, can be verified by means of the condition:

$$\forall x \in \mathbb{X}_s \backslash \mathbb{X}_T, \quad u = K(x) \Rightarrow V(f(x, u)) - \lambda V(x) \leq 0, \ \text{with} \ \lambda \in [0, 1). \tag{5}$$

This means that to have stability $V(x)$ must decrease along the closed-loop trajectory in the annulus.

The sets $\mathbb{X}_s$, $\mathbb{X}_T$, satisfying (5), are (positively) *invariant*. If they are inside constraints, i.e. $\mathbb{X}_s \subseteq \mathbb{X}$, then they are *safe*. For a valid Lyapunov function $V$, the outer safe-set can be defined as a level-set:

$$\mathbb{X}_s = \{x \in \mathbb{X} : V(x) \leq l_s\}. \tag{6}$$

For further definitions, we refer the reader to Blanchini & Miani (2007); Kerrigan (2000). If condition (5), holds everywhere in $\mathbb{X}_s$, then the origin is a stable equilibrium ($X_T = \{0\}$). If (most likely) this holds only outside a non-empty inner set, $\mathbb{X}_T = \{x \in \mathbb{X} : V(x) \leq l_T\} \subset \mathbb{X}_s$, with $\mathbb{X}_T \supset \{0\}$, then the system converges to a neighborhood of the origin and remains there in the future.

**Approach Rationale**    We aim to match or enlarge the safe region of an unknown controller, $K_i(x)$. For a perfect model, $f$, and a safe set $\mathbb{X}_s^{(i)}$, there exists an $\alpha \gg 1$, such that the one-step MPC:

$$K(x) = \underset{u \in \mathbb{U}, \ f(x,u) \in \mathbb{X}_s^{(i)}}{\arg\min} \alpha V(f(x, u)) + \ell(x, u), \tag{7}$$

results in a new safe set, $\mathbb{X}_s^{(i+1)} = \mathcal{C}(\mathbb{X}_s^{(i)})$, the one-step controllable set of $\mathbb{X}_s^{(i)}$ and the feasible region of (7), $\mathbb{X}_s^{(i+1)} \supseteq \mathbb{X}_s^{(i)}$. We soften the state constraints in (7) and use it recursively to estimate $\mathbb{X}_s^{(j)}$, $j > i$. We formulate an algorithm that learns the parameter $\alpha$ as well as the safe set. We train a neural network via SGD to approximate $V$, hence the ROA estimate will not always increase through iterations. To aim for max ROA and minimum MPC horizon, we use cross-validation and verification. We motivate our work by extending theoretical results on MPC stability and a sub-optimality bound for approximate $f$ and $V$. Finally, we provide an error bound on the learned $f$ for having stability.

**Learning and Safety Verification**    We wish to learn $V(x)$ and $\mathbb{X}_s$ from one-step on-policy rollouts, as well as a forward model $\hat{f}(x, u)$. After the learning, the level $l_s$ defining set $\mathbb{X}_s$ will be refined and its safety formally verified a posteriori. This is done by evaluating (5) using the model, starting from a large set of states sampled uniformly within $\mathbb{X}_s \backslash \mathbb{X}_T$. We progressively decrease the level $l_s$ starting from the learned one, and also increase the inner level $l_T$, starting from zero, such that condition (5) holds for $n \geq n_s$ samples in $\mathbb{X}_s \backslash \mathbb{X}_T$. The number of verifying samples, $n_s$, provides a probability lower bound on safety, namely $\mathcal{P}_{safe}(\mathbb{X}_s \backslash \mathbb{X}_T) \geq p(n_s)$, as detailed in (Bobiti & Lazar, 2016). The algorithm is detailed in appendix and based on (Bobiti & Lazar, 2016). For our theoretical results, we assume the search terminates with $\mathcal{P}_{safe}(\mathbb{X}_s \backslash \mathbb{X}_T) \approx 1$ and consider the condition deterministic.

**NN-based dynamics model**    In some MPC applications, it might not be possible to gather sufficient data from demonstrations in order to be able to learn a model that predicts over long sequences. One-step or few-steps dynamics learning based on NNs can suffer when the model is unrolled for longer time. For instance, errors can accumulate through the horizon due to small instabilities either from the physical system or as artifacts from short sequence learning. Although some mitigations are possible for specific architectures or longer sequences (Armenio et al., 2019; Doan et al., 2019; Pathak et al., 2017; Ciccone et al., 2018), we formulate our MPC to allow for a very short horizon and unstable dynamics. Since we learn a surrogate NN forward model, $\hat{f}(x, u)$, from one-step trajectories, we will assume this to have a locally bounded one-step-ahead prediction error, $w(t)$, where:

$$w = f(x, u) - \hat{f}(x, u), \quad \|w\|_2 \leq \mu, \ \forall (x, u) \in \tilde{\mathbb{X}} \times \mathbb{U}, \tag{8}$$

for some compact set of states, $\tilde{\mathbb{X}} \supseteq \mathbb{X}$. We also assume that both $f$ and $\hat{f}$ are locally Lipschitz in this set, with constants $L_{fx}$, $L_{fu}$, and $L_{\hat{f}x}$, $L_{\hat{f}u}$ respectively. A conservative value of $\mu$ can be inferred from these constants as the input and state sets are bounded. It can also be estimated from a test set.

## 3    NEURAL LYAPUNOV MPC

In the context of MPC, a function $V$, which satisfies the Lyapunov property (5) for some local controller $K_0$, is instrumental to formally guarantee stability (Mayne et al., 2000; Limon et al., 2003). We use this insight and build a general Lyapunov function terminal cost for our MPC, based on neural networks. We discuss the formulation of the Lyapunov network and the MPC in  Section 3.1 and Section 3.2 respectively.  In order to extend the controller's ROA while maintaining a short prediction horizon, an alternate optimization scheme is proposed to tune the MPC and re-train the Lyapunov NN. We describe this procedure in Section 3.3 and provide a pseudocode in Algorithm 1.

### 3.1    LYAPUNOV NETWORK LEARNING

We use the Lyapunov function network introduced by Gallieri et al. (2019):

$$V(x) = x^T \left( l_\ell I + V_{net}(x)^T V_{net}(x) \right) x, \tag{9}$$

where $V_{net}(x)$ is a (Lipschitz) feedforward network that produces a $n_V \times n_x$ matrix. The scalars $n_V$ and $l_\ell > 0$ are hyper-parameters. It is easy to verify that (9) satisfies the condition mentioned in (4). In our algorithm, we learn the parameters of the network, $V_{net}(x)$, and a safe level, $l_s$. Note that equation (5) allows to learn $V$ from demonstrations without explicitly knowing the current policy.

**Loss function**    Suppose $\mathcal{D}_K$ denotes a set of state-action-transition tuples of the form $(x, u, x^+)$, where $x^+$ is the next state obtained from applying the policy $u = K(x)$. The Lyapunov network is

trained using the following loss:

$$\min_{V_{net},\, l_s}\ \mathbf{E}_{(x,\, u,\, x^+)\in\mathcal{D}_K}\left[\frac{\mathcal{I}_{\mathbb{X}_s}(x)}{\rho}J_s(x,\, u,\, x^+) + J_{vol}(x,\, u,\, x^+)\right],\qquad(10)$$

where,

$$\mathcal{I}_{\mathbb{X}_s}(x) = 0.5\left(\mathrm{sign}\left[l_s - V(x)\right] + 1\right),\ J_s(x,\, u,\, x^+) = \frac{\max[\Delta V(x),\, 0]}{V(x)+\epsilon_V},$$

$$J_{vol}(x,\, u,\, x^+) = \mathrm{sign}\left[-\Delta V(x)\right]\left[l_s - V(x)\right],$$

$$\Delta V(x) = V\left(x^+\right) - \lambda V(x).$$

In (10), $\mathcal{I}_{\mathbb{X}_s}$ is the indicator function for the safe set $\mathbb{X}_s$, which is multiplied to $J_s$, a function that penalises the instability. The term $J_{vol}$ is a classification loss that tries to compute the correct boundary between the stable and unstable points. It is also instrumental in increasing the safe set volume. The scalars $\epsilon_V > 0$, $\lambda \in [0,1)$, and $0 < \rho \ll 1$, are hyper-parameters, where the latter trades off volume for stability (we take $\rho = 10^{-3}$ as in Richards et al. (2018); Gallieri et al. (2019)). To make sure that $\mathbb{X}_s \subseteq \mathbb{X}$, we scale-down the learned $l_s$ a-posteriori. The loss (10) extends the one proposed by Richards et al. (2018) in the sense that we only use one-step transitions, and safe trajectories are not explicitly labeled before training. This loss is then also used to tune the MPC cost scaling factor for $V$, namely, $\alpha \geq 1$, to guarantee stability. This is discussed next.

## 3.2 NEURAL LYAPUNOV MPC

We improve stability of the initial controller, used to collect data, by replacing it with an MPC solving the following input-limited, soft-constrained, discounted optimal control problem:

$$J^{\star}_{\mathrm{MPC}}(x(t)) = \min_{\underline{\mathbf{u}}}\qquad \gamma^N\alpha V(\hat{x}(N)) + \sum_{i=0}^{N-1}\gamma^i\ell(\hat{x}(i),\hat{u}(i)) + \ell_{\mathbb{X}}(s(i))$$

$$\mathrm{s.t.}\qquad \hat{x}(i+1) = \hat{f}(\hat{x}(i),\hat{u}(i)),\qquad(11)$$

$$\hat{x}(i) + s(i) \in \mathbb{X},\ \forall i \in [0,N],$$

$$\ell_{\mathbb{X}}(s) = \eta_1 s^T s + \eta_2\|s\|_1,\ \eta_1 > 0,\ \eta_2 \gg 0,$$

$$\hat{u}(i) \in \mathbb{U},\ \forall i \in [0,N-1],$$

$$\hat{x}(0) = x(t),$$

where $\hat{x}(i)$ and $\hat{u}(i)$ are the predicted state and the input at $i$-steps in the future, $s(i)$ are slack variables, $\underline{\mathbf{u}} = \{u(i)\}_{i=0}^{N-1}$, the stage cost $\ell$ is given by (3), $\gamma \in (0,1]$ is a discount factor, the function $\hat{f}$ is a forward model, the function $V$ is the terminal cost, in our case a Lyapunov NN from (9), scaled by a factor $\alpha \geq 1$ to provide stability, and $x(t)$ is the measured system state at the current time. The penalty $\ell_{\mathbb{X}}$ is used for state constraint violation, see Kerrigan & Maciejowski (2000).

Problem (11) is solved online given the current state $x(t)$; then, the first element of the optimal control sequence, $u^{\star}(0)$, provides the action for the physical system. Then, a new state is measured, and (11) is again solved, in a *receding horizon*. The implementation details are given in Appendix.

**Stability and safety** We extend results from Limon et al. (2003; 2009) to the discounted case and to the $\lambda$-contractive $V$ from (5). In order to prove them, we make use of the uniform continuity of the model, the SQP solution and the terminal cost, $V$, as done by Limon et al. (2009). Consider the set:

$$\Upsilon_{N,\gamma,\alpha} = \left\{x \in \mathbb{R}^{n_x} : J^{\star}_{\mathrm{MPC}}(x) \leq \frac{1-\gamma^N}{1-\gamma}d + \gamma^N\alpha l_s\right\},\quad \mathrm{where}\quad d = \inf_{x\notin\mathbb{X}_s}\ell(x,0).\quad(12)$$

The following are obtained for system (1) in closed loop with the MPC defined by problem (11). Results are stated for $\mathbb{X}_T = \{0\}$. For $\mathbb{X}_T \neq \{0\}$, convergence would occur to a set instead of 0.

**Theorem 1. Stability and robustness** *Assume that $V(x)$ satisfies (5), with $\lambda \in [0,1)$, $\mathbb{X}_T = \{0\}$. Then, given $N \geq 1$, for the MPC (11) there exist a constant $\bar{\alpha} \geq 0$, a discount factor $\bar{\gamma} \in (0,1]$, and a model error bound $\bar{\mu}$ such that, if $\alpha \geq \bar{\alpha}$, $\mu \leq \bar{\mu}$ and $\gamma \geq \bar{\gamma}$, then, $\forall x(0) \in \mathcal{C}(\mathbb{X}_s)$:*

*1. If $N = 1$, $\mu = 0$, then the system is asymptotically stable for any $\gamma > 0$, $\forall x(0) \in \Upsilon_{N,\gamma,\alpha}$.*

2. *If $N > 1$, $\mu = 0$, then the system reaches a set $\mathbb{B}_\gamma$ that is included in $\mathbb{X}_s$. This set increases with decreasing discount factors, $\gamma$, $\forall x(0) \in \Upsilon_{N,\gamma,\alpha}$. $\gamma = 1 \Rightarrow B_\gamma = \{0\}$.*

3. *If $\alpha V(x)$ is the optimal value function in $\mathbb{X}_s$ for the problem, $\mu = 0$, and if $\mathcal{C}(\mathbb{X}_s) \neq \mathbb{X}_s$, then the system is asymptotically stable, $\forall x(0) \in \Upsilon_{N,\gamma,\alpha}$.*

4. *If $\mu = 0$, then $\alpha \geq \bar{\alpha}$ implies that $\alpha V(x) \geq V^\star(x), \forall x \in \mathbb{X}_s$, where $V^\star$ is the optimal value function for the infinite horizon problem with cost (3) and subject to (2).*

5. *The MPC has a stability margin. If the MPC uses a surrogate model satisfying (8), with one-step error bound $\|w\|_2^2 < \bar{\mu}^2 = \frac{1-\lambda}{L_V L_{\hat{f}_x}^{2N}} l_s$, then the system is Input-to-State (practically) Stable (ISpS) and there exists a set $\mathbb{B}_{N,\gamma,\mu}: x(t) \rightarrow \mathbb{B}_{N,\gamma,\mu}$, $\forall x(0) \in \beta \Upsilon_{N,\gamma,\alpha}$, $\beta \leq 1$.*

Theorem 1 states that for a given horizon length $N$ and contraction factor $\lambda$, one can find a minimum scaling of the Lyapunov function $V$ and a lower bound on the discount factor such that the system under the MPC is stable. Hence, if the model is perfect, then the state would converge to the origin as time progresses. If the model is not perfect, then the safety of the system depends on the size of the model error. If this error is less than the maximum tolerable error, $\mu \leq \bar{\mu}$, then the system is *safe*: the state converges to a bound, the size of which increases with the size of the model error, the prediction horizon $N$, and is inversely proportional to $\alpha$. In other words, the longer the predictions with an incorrect model, the worse the outcome. Note that the ROA also improves with larger $\alpha$ and $\gamma$. The proof of the theorem is provided in Appendix. Results hold with the verified probability, $\mathcal{P}_{safe}(\mathbb{X}_s)$.

**Performance with surrogate models** In order to further motivate for the search of a $V$ giving the largest $\mathbb{X}_s$, notice that a larger $\mathbb{X}_s$ can allow for shortening the MPC horizon, yielding the same ROA. Contrary to Lowrey et al. (2018), we demonstrate how model mismatch and longer horizons can decrease performance with respect to an infinite-horizon oracle with same cost and perfect model.

Let $\mathbf{E}_\mathcal{D}[J_{V^\star}(K^\star)]$ define the expected infinite-horizon performance of the optimal policy $K^\star$, evaluated by using the expected infinite-horizon performance (value function), $V^\star$, for the stage cost (3) and subject to (2). Similarly, let $\mathbf{E}_{x \in \mathcal{D}}[J^\star_{\text{MPC}}(x)]$ define the MPC's expected performance with the learned $V$, when a surrogate model is used and $\mathbf{E}_{x \in \mathcal{D}}[J^\star_{\text{MPC}}(x; f)]$ when $f$ is known.

**Theorem 2. Performance** *Assume that the value function error is bounded for all $x$, namely, $\|V^\star(x) - \alpha V(x)\|_2^2 \leq \epsilon$, and that the model error satisfies (8), for some $\mu > 0$. Then, for any $\delta > 0$:*

$$
\begin{aligned}
\mathbf{E}_{x \in \mathcal{D}}[J^\star_{MPC}(x)] - \mathbf{E}_{x \in \mathcal{D}}[J^\star_{V^\star}(x)] \quad &\leq \frac{2\gamma^N \epsilon}{1-\gamma^N} + \left(1 + \frac{1}{\delta}\right) \|Q\|_2 \sum_{i=0}^{N-1} \gamma^i \left(\sum_{j=0}^{i-1} \bar{L}_f^j\right)^2 \mu^2 \\
&+ \left(1 + \frac{1}{\delta}\right) \gamma^N \alpha L_V \left(\sum_{i=0}^{N-1} \bar{L}_f^i\right)^2 \mu^2 + \bar{\psi}(\mu) \\
&+ \delta \, \mathbf{E}_{x \in \mathcal{D}}\left[J^\star_{MPC}(x; f)\right],
\end{aligned}
$$

*where $\bar{L}_f = \min(L_{\hat{f}_x}, L_{f_x})$ and $\bar{\psi}$ is a $\mathcal{K}_\infty$-function representing the constraint penalty terms.*

Theorem 2 is related to Asadi et al. (2018) for value-based RL. However, here we do not constrain the system and model to be stable, nor assume the MPC optimal cost to be Lipschitz. Theorem 2 shows that a discount $\gamma$ or a shorter horizon $N$ can mitigate model errors. Since $\gamma \ll 1$ can limit stability (Theorem 1) we opt for the shortest horizon, hence $N = 1$, $\gamma = 1$. Proof of Theorem 2 is in Appendix.

**MPC auto-tuning** The stability bounds discussed in Theorem 1 can be conservative and their computation is non-trivial. Theoretically, the bigger the $\alpha$ the larger is the ROA (the safe region) for the MPC, up to its maximum extent. Practically, for a very high $\alpha$, the MPC solver may not converge due to ill-conditioning. x Initially, by using the tool from Agrawal et al. (2019) within an SQP scheme, we tried to tune the parameters through gradient-based optimization of the loss (10). These attempts were not successful, as expected from the considerations in Amos et al. (2018). Therefore, for practical reasons, in this work, we perform a grid search over the MPC parameter $\alpha$. Note that the discount factor $\gamma$ is mainly introduced for Theorem 2 and analysed in Theorem 1 to allow for future combination of stable MPC with value iteration.

### 3.3 Learning algorithm

Our alternate optimization of the Lyapunov NN, $V(x)$, and the controller is similar to Gallieri et al. (2019). However, instead of training a NN policy, we tune the scaling $\alpha$ and learn $V(x)$ used by the MPC (11). Further, we extend their approach by using a dataset of demonstrations, $\mathcal{D}_{\text{demo}}$, instead of an explicitly defined initial policy. These are one-step transition tuples, $(x(0), u(0), x(1))_m$, $m = 1, \ldots, M$, generated by a (possibly sub-optimal) stabilizing policy, $K_0$. Unlike in the approach by Richards et al. (2018), our $V$ is a piece-wise quadratic, and it is learned without labels. We in fact produce our own psuedo-labels using the sign of $\Delta V(x)$ in (10) in order to estimate $l_s$. The latter means that we don't require episode-terminating (long) rollouts, which aren't always available from data nor accurate when using a surrogate. Also, there is no ambiguity on how to label rollouts.

---

**Algorithm 1** Neural Lyapunov MPC learning

**In:** $\mathcal{D}_{\text{demo}}, \hat{f}, \lambda \in [0, 1), \{l_\ell, \epsilon_{\text{ext}}\} > 0, \gamma \in (0, 1], N \geq 1, \alpha_{\text{list}},$
$\quad N_{ext}, N_V, \epsilon_V, V_{init}, \ell(x, u)$
**Out:** $V_{net}, l_s, \alpha^\star$

---

$\mathcal{D} \leftarrow \mathcal{D}_{\text{demo}}$
$V_{net} \leftarrow V_{init}$
**for** $j = 0 \ldots N_V$ **do**
$\quad | \quad (V_{net}, l_s, \mathbb{X}_s) \leftarrow$ Adam step on (10)
**end**
**for** $i = 0 \ldots N_{ext}$ **do**
$\quad \mathcal{D} \leftarrow \mathcal{D}_{\text{demo}} \cap (1 + \epsilon_{\text{ext}})\mathbb{X}_s$
$\quad$ **for** $\alpha \in \alpha_{list}$ **do**
$\quad\quad | \quad \mathcal{U}_1^\star \leftarrow \texttt{MPC}(V_{net}, \hat{f}, \mathcal{D}; \alpha), \text{ from (11)}$
$\quad\quad | \quad \mathcal{D}_{\text{MPC}}(\alpha) \leftarrow \texttt{one\_step\_sim}(\hat{f}, \mathcal{D}, \mathcal{U}_1^\star)$
$\quad\quad | \quad \mathcal{L}(\alpha) \leftarrow$ Evaluate (10) on $\mathcal{D}_{\text{MPC}}(\alpha)$
$\quad$ **end**
$\quad \alpha^\star \leftarrow \arg\min(\mathcal{L}(\alpha))$
$\quad \mathcal{D} \leftarrow \mathcal{D}_{\text{MPC}}(\alpha^\star)$
$\quad V_{net} \leftarrow V_{init}$
$\quad$ **for** $j = 0 \ldots N_V$ **do**
$\quad\quad | \quad (V_{net}, l_s, \mathbb{X}_s) \leftarrow$ Adam step on (10)
$\quad$ **end**
**end**

---

Once the initial $V$, $\mathbb{X}_s$ are learned from the demonstrations, we use $V$ and a learned model, $\hat{f}$, within the MPC. We tune the MPC parameter $\alpha$ to minimize the loss defined in (10), using $(1 + \epsilon_{\text{ext}})\mathbb{X}_s$ as a new *enlarged* target safe set instead of $\mathbb{X}_s$. This is done to push the safe set to extend. We propose Algorithm 1, which runs multiple iterations where after each of them the tuned MPC serves as a demonstrator for training the next $V$ and $\mathbb{X}_s$ to verify the MPC in closed-loop with the model. Since it is not guaranteed that the ROA will increase during learning, we select the Lyapunov function and the MPC using the criteria that the percentage of stable points ($\Delta V < 0$) increases and that of unstable points decreases while iterating over $j$ and $i$ when evaluated on a validation set.

In Algorithm 1, MPC denotes the proposed Neural Lyapunov MPC, while `one_step_sim` denotes a one-step propagation of the MPC action into the system surrogate model. To train the parameters of $V$ and the level-set $l_s$, Adam optimizer is used Kingma & Ba (2014). A grid search over the MPC parameter $\alpha$ is performed. A thorough tuning of all MPC parameters is also possible, for instance, by using black-box optimisation methods. This is left for future work.

## 4 Numerical experiments

Through our experiments, we show the following: 1) increase in the safe set for the learned controller by using our proposed alternate learning algorithm, 2) robustness of the one-step NLMPC compared to a longer horizon MPC (used as demonstrator) when surrogate model is used for predictions, and 3) effectiveness of our proposed NLMPC against the demonstrator and various RL baselines.

**Constrained inverted pendulum** In this task, the pendulum starts near the unstable equilibrium ($\theta = 0°$). The goal is to stay upright. We bound the input so that the system cannot be stabilized if $|\theta| > 60°$. We use an MPC with horizon 4 as a demonstrator, with terminal cost, $500 x^T P_{\text{LQR}} x$,

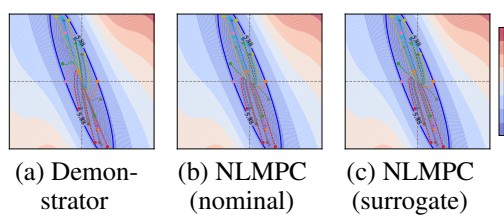

(a) Demon-strator    (b) NLMPC (nominal)    (c) NLMPC (surrogate)

Figure 1: **Inverted Pendulum: Testing learned controller on nominal system.** Lyapunov function with safe trajectories. NLMPC learns successfully and also transfers to surrogate model.

Table 1: **Inverted Pendulum: Learning on nominal model.** With iterations, the number of points verified by the controller increases.

| Iter. | Loss $\log(1+x)$ | Verif. % | Not Verif. % |
|---|---|---|---|
| 1 | 3.21 | 13.25 | 0.00 |
| 2 | 1.08 | 13.54 | 0.00 |

Table 2: **Car: Learning on nominal model.**

| ITER. | LOSS | VERIF. | NOT VERIF. |
|---|---|---|---|
| 1 | 1.55 | 92.20 | 4.42 |
| 2 | 0.87 | 93.17 | 4.89 |
| 3 | 0.48 | 94.87 | 3.89 |

Table 3: **Car: Learning on surrogate model.**

| ITER. | LOSS | VERIF. | NOT VERIF. |
|---|---|---|---|
| 1 | 1.84 | 91.74 | 8.26 |
| 2 | 1.43 | 92.26 | 7.74 |
| 3 | 1.65 | 91.61 | 8.39 |

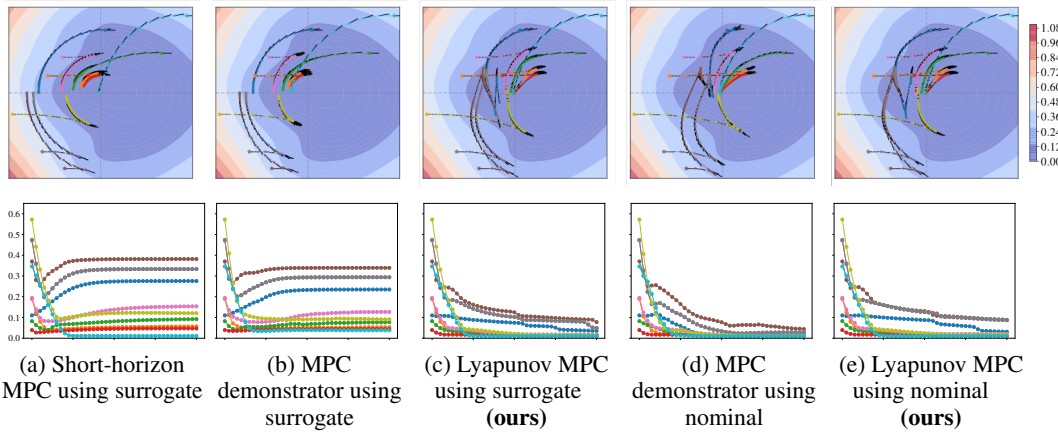

| (a) Short-horizon MPC using surrogate | (b) MPC demonstrator using surrogate | (c) Lyapunov MPC using surrogate **(ours)** | (d) MPC demonstrator using nominal | (e) Lyapunov MPC using nominal **(ours)** |

Figure 2: **Car kinematics: Transfer from surrogate to a nominal model. Top**: Lyapunov function contours at $\phi = 0$ with trajectories for 40 steps. **Bottom**: Lyapunov function evaluated for specific policy on several initial states (decreasing means more stable).

where $P_{\mathrm{LQR}}$ is the LQR optimal cost matrix. This is evaluated on $10K$ equally spaced initial states to generate the dataset $\mathcal{D}_{\mathrm{demo}}$. We train a grey-box NN model, $\hat{f}$ using $10K$ random transition tuples. More details are in Appendix. The learned $V$ and $\alpha$, obtained from Algorithm 1, produce a one-step MPC that stabilizes both the surrogate and the actual system. Table 1 shows that the loss and percentage of verified points improve across iterations. The final ROA estimate is nearly maximal and is depicted along with the safe trajectories, produced by the MPC while using predictions from the nominal and surrogate models, in Figure 1. The performance matches that of the baseline and the transfer is successful due to the accuracy of the learned model. A full ablation study is in Appendix.

**Constrained car kinematics** The goal is to steer the car to the $(0, 0)$ position with zero orientation. This is only possible through non-linear control. The vehicle cannot move sideways, hence policies such as LQR is not usable to generate demonstrations. Thus to create $D_{\mathrm{demo}}$, an MPC with horizon 5 is evaluated over $10K$ random initial states. The surrogate, $\hat{f}$ is a grey-box NN trained using $10K$ random transition tuples. More details are in Appendix. Figure 3 shows the learning curves, training of the Lyapunov function over iterations and line-search for the MPC auto-tuning. Table 2 summarises the metrics improvement across the iterations, indicating an increase in the ROA when a perfect model is used. With an imperfect model, the second iteration gives the best results, as shown in Table 3.

We test the transfer capability of the approach in two ways. First, we learn using the nominal model and test using the surrogate model for the MPC predictions. This is reported in Appendix for the sake of space. Second, the learning is performed using the surrogate model as in Algorithm 1, and the MPC is then tested on the nominal model while still using the surrogate for prediction. This is depicted in Figure 2. Our MPC works better than the demonstrator when using the incorrect model. The learned MPC transfers successfully and completes the task safely.

**Comparison to baselines** Prior works such as constrained policy optimization (CPO) (Achiam et al., 2017) provide safety guarantees in terms of constraint satisfaction that hold in expectation. However, due to unavailability of a working implementation, we are unable to compare our approach against it. Instead to enforce safety constraints during training of the RL algorithms, we use two different strategies: v1) early episode termination; v2) reward shaping with a constraint penalty. The v2 formulation is similar to the one used in Ray et al. (2019), which demonstrated its practical equivalence to CPO when tuned. We compare our approach against model-free and model-based

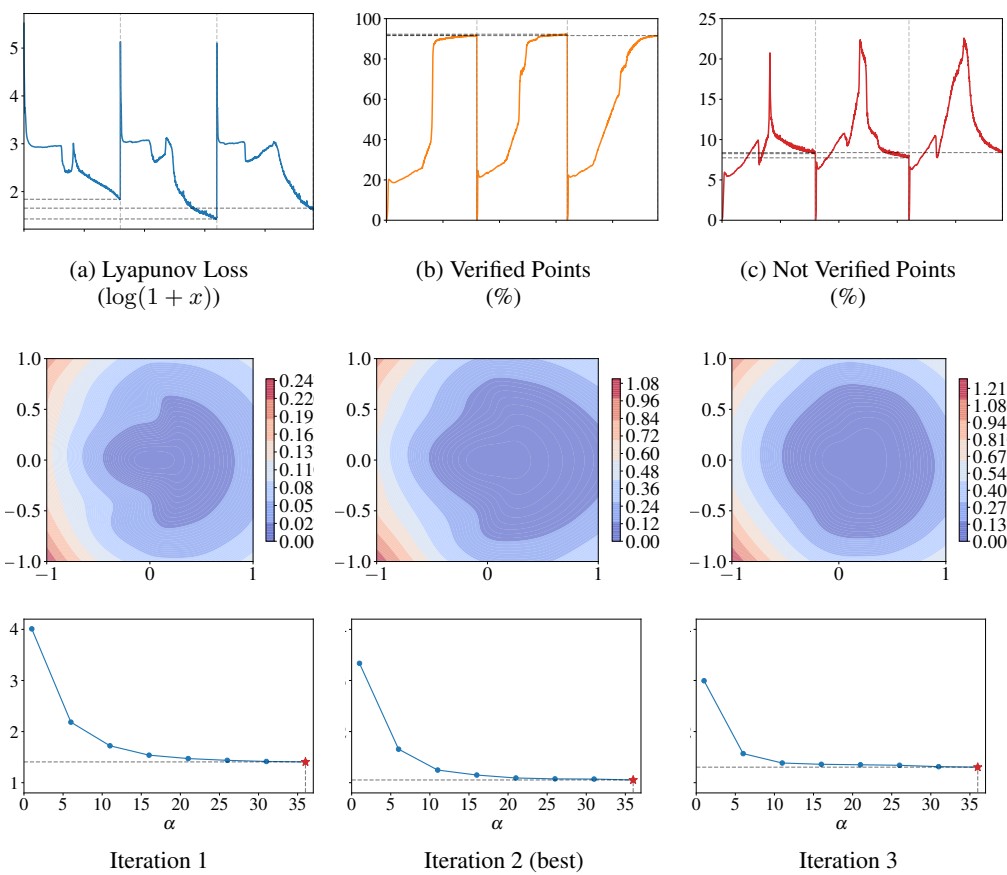

Figure 3: **Car kinematics: Alternate learning on surrogate model.** After every $N_V = 800$ epochs of Lyapunov learning, the learned Lyapunov function is used to tune the MPC parameters. **Top:** The training curves for Lyapunov function. Vertical lines separate iterations. **Middle:** The resulting Lyapunov function $V$ at $\phi = 0$ with the best performance. **Bottom:** Line-search for the MPC parameter $\alpha$ to minimize the Lyapunov loss (10) with $V$ as terminal cost. The loss is plotted on the y-axis in a $\log(1 + x)$ scale. The point marked in red is the parameter which minimizes the loss.

baseline algorithms. For the model-free baselines, we consider the on-policy algorithm proximal policy optimization (PPO) (Schulman et al., 2017) and the off-policy algorithm soft actor-critic (SAC) (Haarnoja et al., 2018). For model-based baselines, we consider model-based policy optimization (MBPO) (Janner et al., 2019) and the demonstrator MPC. Further details about the reward shaping and learning curves are in Appendix.

We consider the performance of learned controllers in terms stability and safety. Stability performance is analogous to the success rate in performing the set-point tracking task. We consider a task is completed when $||x(T)||_2 < 0.2$ where $T$ is the final time of the trajectory. For the car, we exclude the orientation from this index. The safety performance combines the former with state constraints satisfaction over the entire trajectory. As shown in Table 4, for the inverted pendulum, all the policies lead to some safe trajectories. Note that the demonstrator (which has an LQR terminal cost) is an optimal controller and is the maximum performance that can be achieved. In terms of stability performance, our approach performs as good as the demonstrator MPC. The RL trained policies give sub-optimal behaviors, i.e. sometimes the system goes to the other equilibria. For the car, the demonstrator MPC is a sub-optimal policy. NLMPC improves upon it in performance and it is on par with it in terms of safety. NLMPC also significantly outperforms all of the considered RL baselines while using lesser number of samples for learning[1].

---

[1]For all our experiments, training datapoints: PPO: $4 \times 10^6$, SAC: $4 \times 10^6$, MBPO: $2.4 \times 10^5$, NLMPC: $10^4$ (random) + $10^4$ (demonstrations).

Table 4: **Comparison with baselines.** We compare our NLMPC (with surrogate model for predictions) with baselines. In the pendulum, our approach is second to the demonstrator for less than $1\%$ margin. In the car task, NLMPC performs better than all baselines and improves convergence from the demonstrator, while it is nearly on par with the latter on constraints.

| ALGORITHM | CONSTRAINED INVERTED PENDULUM | | CONSTRAINED CAR KINEMATICS | |
|---|---|---|---|---|
| | STABILITY (%) | SAFETY (%) | STABILITY (%) | SAFETY (%) |
| PPO (v1) | 14.67 | 14.66 | 15.17 | 0.50 |
| PPO (v2) | 26.33 | 26.33 | 8.16 | 0.83 |
| SAC (v1) | 29.99 | 29.99 | 12.33 | 0.00 |
| SAC (v2) | 27.17 | 27.17 | 8.00 | 0.00 |
| MBPO (v1) | 12.67 | 12.67 | 6.00 | 0.00 |
| MBPO (v2) | 26.00 | 26.00 | 6.00 | 0.00 |
| MPC (DEMO) | **36.00** | **36.00** | 81.33 | **73.33** |
| **NLMPC** | 35.33 | 35.33 | **86.00** | 72.67 |

## 5 RELATED WORK

Stability and robustness of MPC and of discounted optimal control have been studied in several prior works Mayne et al. (2000); Rawlings & Mayne (2009); Limon et al. (2009; 2003); Raković et al. (2012); Gaitsgory et al. (2015). Numerical stability verification was studied in Bobiti (2017); Bobiti & Lazar (2016) and, using neural network Lyapunov functions in Berkenkamp et al. (2017); Gallieri et al. (2019). Neural Lyapunov controllers were also trained in Chang et al. (2019). MPC solvers based on iterative LQR (iLQR) were introduced in Tassa et al. (2012). Sequential Quadratic Program (SQP) was studied in Nocedal & Wright (2006). NNs with structural priors have been studied in Quaglino et al. (2020); Yıldız et al. (2019); Pozzoli et al. (2019). Value functions for planning were learned in Lowrey et al. (2018); Deits et al. (2019); Buckman et al. (2018). Gallieri et al. (2019) learned a NN Lyapunov function and an NN policy with an alternating descent method, initialized using a known stabilizing policy. We remove this assumption and use MPC. Suboptimality was analysed in Grune & Rantzer (2008) for MPC and in Janner et al. (2019) for policies. Differently from NNs, non-parametric models have been largely studied for control, see for instance Koller et al. (2018); Hewing et al. (2020) and references therein for closed-form results using Gaussian processes.

## 6 CONCLUSIONS

We presented Neural Lyapunov MPC, a framework to train a stabilizing non-linear MPC based on learned neural network terminal cost and surrogate model. After extending existing theoretical results for MPC and value-based reinforcement learning, we have demonstrated that the proposed framework can incrementally increase the stability region of the MPC through offline RL and then safely transfer on simulated constrained non-linear control scenarios. Through comparison of our approach with existing RL baselines, we showed how NNs can be leveraged to achieve policies that perform at par with these methods while also having provable safety guarantees.

Future work could address the reduction of the proposed sub-optimality bound, for instance through the integration of value learning with Lyapunov function learning as well as the optimal selection of the MPC prediction horizon. A broader class of stage costs and rewards could also be investigated.

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
