# OpenReview forum: "Neural Lyapunov Model Predictive Control"
_ICLR.cc/2021/Conference — Reject_

### Official Review · AnonReviewer4 · 2020-10-25
**Promising work, but unclear theoretical novelty and empirical achievements**

**Rating:** 7
**Confidence:** 3

**Review:**

This paper addresses the question of how to stabilize a system in a vicinity of an equilibrium. While the majority of reinforcement learning algorithms rely on trial and error, which may damage the system, the authors introduce an algorithm for safe exploration and control. A traditional approach in model-based RL is to use MPC with a surrogate forward model to minimize a planning objective comprising a sum of stage costs along with a terminal cost, often chosen as an approximated value function -i.e. the optimal expected cost-to-go- which can be learned by a Bellman equation. Instead, this work is placed in the framework of Robust MPC, where this value function is replaced by a Luyapunov function $V$, which is related to the notion of stability and is only constrained to decrease along trajectories. Such a Luyapunov function, when available, provides both a safe region, defined as a level-set of V, and a MPC policy for which stability analyses have been developed: the authors extend a result from Limon et al. (2003; 2009) to show that this MPC policy enjoys asymptotic stability in general, and input-to-state stability in the presence of small enough model errors. Accordingly, the authors propose a scheme allowing to learn a Lyapunov function $V$ from demonstration data only, through a loss function that penalizes increments of $V$ along one-step transitions. A regularization parameter $\alpha$ of this MPC, which balances stability with constraints satisfaction and stage costs, is also learned jointly by an alternative training procedure. This approach is evaluated empirically on two standard constrained non-linear continuous control tasks.


Strong points:
1. This paper is clearly written, well motivated, honest about its place in the literature, and all derivations seem technically correct.
2. By bringing together existing results and techniques (MPC stability analyses, value-based RL analyses, LuyapunovNet), the authors manage to relax several assumptions of prior works (no need for access to the perfect dynamics or a stabilizing policy, but only to a demonstration dataset) which makes the approach more practical.
3. Empirically, the learned Luyapunov functions seem to effectively capture useful stability information, since the proposed approach outperforms a standard MPC with a longer planning horizon. Even better, this observation is theoretically justified by Lemma 2: errors of the surrogate model compound when used in an MPC, which is detrimental for long-term planning. Conversely, if $V$ contains long-term information, it can directly be used for short-sighted planning, similarly to being greedy with respect to a value function.


Weak points:
1. The part which was the least clear to me is the *Performance with surrogate models* paragraph, with Lemma 2. The authors draw a parallel between Luyapunov functions in control theory and value functions in RL, but the latter are not really defined clearly in the text. The authors state in their introduction that they treat "the learned Lyapunov NN as an estimate of the value function" and later they mention a "correct" value function $V^*$ for optimal "expected infinite-horizon performance", but this quantity is nowhere defined. I suppose $V^*$ is an expected infinite sum of discounted stage costs, but which costs? The same as in equation (3)? If so, I find it hard to believe that the assumption of Lemma 2 should be satisfied ($V^*$ close to $\alpha V$), given that the stage cost $l$ of (3) used to define $V^*$ does not appear in the loss (10) used to learn the Lyapunov function $V$.
2. I find it difficult to assess the novelty of the theoretical results. The authors are honest in stating that they are extending/adapting known results, but it is not precisely stated what their added value is. Moreover, the abstract mentions that "we also present theorems" but in the article these results are presented as lemmas, which to me usually suggests that they are either instrumental to another result (which they are not), or of minor importance.
3. One of the main claim of the paper is the ability of the method to expand a demonstrated safe region. However, this is not really observed consistently across tasks and iterations. For instance, it is not the case in Figure 3. Likewise, the Table 1 states that "With iterations, the number of points verified by the controller increases", but only a two iterations is provided (i.e. a single opportunity to increase/decrease). This seems a bit suspicious and suggests that the ratio of verified points may actually decrease on subsequent iterations, as it does on iteration 3 of Table 3.


To conclude, I lean toward recommending acceptance, but I am ready to increase my score provided that the authors improve the clarity on both the analogy between Lyapunov and value functions, and state more clearly the novelty of their theoretical contributions.


Minor remarks and typos:
* I do not really see the relevance of the safety performance metric in the Inverted Pendulum experiment (Fig. 3): the state constraints are loose enough that every successful trajectory is also considered safe, so this safety plot (right) does not really bring any new information to the table.
* p2, Equation (5): X_s\ **X_T**
* p3, Learning and Safety Ver**i**fication
* p6, to minimize the loss defined in **(11)**

---

> ### Author Response · Authors · 2020-11-17
> **Reviewer4 Response Part 1- Thank you for the feedback**
>
> We thank the reviewer for thoroughly reading the paper and the positive and constructive feedback. We particularly appreciate that our work was considered “clear”, “well-motivated”, “honest”, “technically correct” and its relevance was recognized. We have tried to address all the concerns by the reviewer in the following but are open to further discussions. We will fix the typos and make the required additions to the revised upload of the paper. We hope we convince the reviewer to possibly further improve their score.
>
> Please find our comments on the specific questions/concerns that have been raised:
>
>
> #### 1. Regarding performance with surrogate models
>
>
> Yes, by using the Lyapunov function as the terminal cost, we want to approximate the tail of an infinite-horizon control formulation that uses the stage cost defined in (3). We do not expect the error to be small with respect to the correct value function. The error can, however, be upper-bounded since all considered functions, the system dynamics, are Lipschitz and the constraints sets are bounded. Lemma 2 aims to break down the different components that affect performance and communicate the role of horizon length in the presence of model uncertainty. It would be possible in principle to combine the approach with value estimation to reduce the error but this is beyond the scope of the paper. The reason for not using the stage cost directly is that the unknown controller might not be optimal with respect to this stage cost (reward) but it is generally an engineered working solution. This choice was highly beneficial for instance in the car scenario, where our initial long horizon MPC used to generate the data produced suboptimal solutions due to the optimizer tolerances, correct decrease rate cannot be guaranteed, differently from an LQR. Moreover, the long horizon MPC trajectories converge only to a neighborhood of the origin for the car. Despite this, our algorithm learns a Lyapunov function that can provide the system to even outperform the initial MPC, as kindly pointed out by the reviewer, in terms of convergence to a smaller neighborhood.
>
> #### 2. Regarding the novelty of the theoretical results
>
> In the revised paper, we will rectify calling the theoretical results as "lemmas" instead of "theorems".
>
> We mainly extend existing results in Theorem 1 by considering the discount factor in the formulation and the contraction factor in the Lyapunov inequality instead of the loss. For the rest of Theorem 1, we build and proceed upon the same steps of the cited articles from Limon et al. (2003, 2009). As also remarked by the reviewer, we state the same on page 4 in the paragraph on “stability and safety”.  We did not mention the contraction instead of the loss but we will make this clearer in the revised upload.
>
> Overall, Theorem 1 aims to motivate our proposed algorithm, which is our main contribution in this work. We thought it was important, however, to state the theorem to mark the relevance of learning a Lyapunov function instead of just a value function surrogate. In Theorem 2, we instead aim to close the gap with Lowrey et al. (2018) in terms of performance evaluation. This is also useful to understand the contribution of the horizon length and model error. With a perfect model, a longer horizon is indeed beneficial as claimed by Lowrey et al. (2018), however, this is true only for a perfect model. In this case, it is also better in terms of the size of the stable region of the MPC. However, having a Lyapunov function that induces a larger safe set allows one to employ a shorter horizon and retain the same stable region with a shorter horizon. We reinforce the statement by demonstrating that, with function approximation in the model, a shorter horizon can become much more beneficial also in terms of optimality. The bound in Theorem 2 can be used to estimate the effect on convergence by model error and horizon length. We will clarify this further in the revised upload.

---

> > ### Author Response · Authors · 2020-11-17
> > **Reviewer4 Response Part 2**
> >
> > #### 3. Regarding the expansion of the demonstrator safe region
> >
> > As stated on page 2, the increase in the ROA is possible if the model is perfect, the MPC solver is perfect (K=argmin), and the Lyapunov function and the safe set are known exactly. In practice, the presence of function approximation limits the achievable results. That’s why on page 2 we also state that “we aim to match or enlarge” the ROA of the unknown demonstrator. Further, we add to it that “assumptions are relaxed and results are presented for the use of function approximation”. The results in Theorems 1 and 2 don’t claim that we increase the ROA using our proposed algorithm. Instead, we claim that the ROA exists for a given discount, terminal penalty $\alpha$, and model error bound. We have updated Theorem 1 with a computation of the model error bound. For these reasons, the proposed algorithm cannot guarantee that the size of the stable region will always increase. We try to enforce that by introducing the factor $(1+\epsilon)X_s$ in the loss for the MPC parameter $\alpha$, which aims to extend $X_s$ by a factor $\epsilon$ (like 10%) at each iteration. This is conditioned to the result of the $\alpha$ search as well as the training of $V$. While in theory the optimal value for the parameter $\alpha$ can be as large as possible and will lead to a ROA increase, in practice, there is a limit beyond which the problem can become ill-conditioned for a particular MPC solver. That is why we perform the search only within a fixed range $[0, a_{\textrm{max}}]$, as mentioned in the paragraph on “MPC auto-tuning” on page 5.
> >
> > Therefore, in practice, yes, the ratio of verified points can decrease over iterations. This is also why we perform cross-validation on our Lyapunov function and we choose the best iterations. We performed only three iterations due to limited computation but reported all of them. As can be seen from Table 1, for the pendulum, two iterations reached the largest achievable result, after which it is the same. However, for the car example, it does not improve at all times. The third iteration in Table 3 shows a decrease in the verifiable region when the surrogate model is used. This is not the case when a perfect model is used, which seems to confirm the limits from function approximation. We always pick the best iteration from these tables as well as cross-validating the Lyapunov function.  We will add this explanation to the revised upload of the paper.
> >
> >
> > References
> >
> > 1. Limon, D., T. Alamo, and E. F. Camacho. "Stable constrained MPC without terminal constraint." Proceedings of the 2003 American Control Conference, 2003.. Vol. 6. IEEE, 2003.
> > 2. Limon, D., et al. "Input-to-state stability: a unifying framework for robust model predictive control." Nonlinear model predictive control. Springer, Berlin, Heidelberg, 2009. 1-26.

---

> > > ### Comment · AnonReviewer4 · 2020-11-21
> > > **Thank you for your answers**
> > >
> > > 1. Performance with surrogate models.
> > >
> > > I am satisfied with the authors' answer, and I now find the revised version of the paragraph much clearer.
> > > I think there only remains a typo in "For a task specified by the stage cost in (10)", where (11) or (3) was probably meant? ((10) does not contain any stage cost)
> > > I now understand that Theorem 2 effectively bounds the MPC suboptimality with respect to the model error $\mu$, true/surrogate dynamics smoothness $L_f$, and the error in the terminal cost $V$ used for planning with respect to $V^\star$. I also appreciate that the proof provided in Appendix A is quite easy to read.
> > >
> > > However, it is not obvious to me why the authors state that "Theorem 2 shows that a discount $\gamma$ or a shorter horizon $N$ can mitigate model errors". Indeed, considering only the terms in the bound which depend on $\mu$, it is true that the second term strictly increase with $N$, but the third term is in $\gamma^N(1-L_f^N)$ which is not monotonous with $N$ in general, and converges to $0$ as $N\to\infty$ if $L_f<1/\gamma$.
> > >
> > > And beside model errors, increasing $N$ also helps mitigating the value function error $\epsilon$ (first term), which has every reason to be large since, as the authors confirmed, $V$ learned by (5) has no particular reason to be close to $V^\star$.
> > >
> > > In any case, I think that this result constitutes an interesting research perspective for further work, e.g. to derive an algorithm to adaptively select the optimal horizon N which adequately balances value estimation errors and model errors, if these can be reliably estimated.
> > >
> > >
> > > 2. Novelty of the theoretical results
> > >
> > > I thank the authors for clarifying their claimed contributions, beyond the original work of Limon et al. (2003, 2009). I understand that they mainly adjust the setting (discount, $\lambda$-contraction of $V$) and then follow the original proofs. This seems quite incremental, but it is not necessarily problematic, especially since this theoretical result provides practical insights on the relationship between the scaling of $V$, the size of the associated ROA, and the interplay of model errors and the prediction horizon, which sheds light on the experimental results.
> > >
> > > 3. Regarding the expansion of the demonstrator safe region
> > >
> > > I understand that resorting to function approximation can degrade results compared to the theoretical setting, and I appreciate that this is mitigated by the ability to perform online safety verifications by means of the cross-validation procedure. I initially thought that the authors claimed that the ROA was guaranteed to increase, but I now have a better understanding that Theorem 1 rather provides the existence of the ROA, and that the attempt of ROA expansion is more experimental: leverage theoretival insights and iteratively increase the MPC parameter $\alpha$ with the hope of increasing $X_s$.
> > >
> > >
> > > The authors have properly addressed my concerns, and I have updated my score to reflect this.

---

> > > > ### Author Response · Authors · 2020-11-23
> > > > **Thank you for increasing the score and for the additional feedback**
> > > >
> > > > We would like to thanks the reviewer for increasing their score and for their relevant feedback.
> > > >
> > > > The reviewer is correct about the typo (we will correct as "with cost (3) subject to constraints (2)") and the third term in our bound being not monotonic. We will correct the claim. It was indeed the second term in the bound that motivated us for the choice of a short horizon. It is also correct and a good suggestion that this result could serve to motivate the development of a method for selecting the horizon length; for instance if an estimate is provided for the model and the value errors. We also believe this could be performed in future work. We also think the approach could be extended in the future by complementing the Lyapunov loss with a loss used for value function learning to possibly mitigate the value estimation error term. We have not investigate this yet because we wanted to demonstrate the Lyapunov function part but it appears of interest. This also reconnects to the points raised by Reviewer 1. We will propose these items for future work in the additional space. We will make these amendments as soon as possible and upload a further revision.

---

### Official Review · AnonReviewer1 · 2020-10-29
**Poorly motivated and unclear contributions**

**Rating:** 3
**Confidence:** 4

**Review:**

This paper proposes an MPC algorithm based on a learned (neural network) Lyapunov function. In particular, they learn both the Lyapunov function and the forward model of the dynamics, and then control the system using an MPC with respect to these models.

Cons
- Poorly written
- Unclear connections to related work
- Weak experiments

It is unclear exactly what problem the authors are attempting to solve. In general, the authors introduce a large amount of notation and theory, but very little of it appears to be directly related to their algorithm. For example, they refer to the stability guarantees afforded by Lyapunov functions, but as far as I can tell, they never prove that their algorithm actually learns a Lyapunov function (indeed, Lemma 1 starts with “Assume that V(x) satisfies (5) [the Lyapunov condition] ...”).

Similarly, they allude to “robustness margins to model errors”, but nothing in the algorithm actually takes into account model errors. Is the point of these margins just to show that they exist? If so, it’s not clear the results (either theoretical or empirical) are very meaningful, given that they depend on the unknown model error (which they assume to be bounded).

In addition, the different loss functions they use (e.g., (10)) are poorly justified. Why is this loss the right one to use to learn a Lyapunov function?

Furthermore, the authors’ approach is closely related to learning the value function and planning over some horizon using the value function as the terminal cost (indeed, the value function is a valid Lyapunov function, but not necessarily vice versa). For instance;

Buckman et al., Sample-Efficient Reinforcement Learning with Stochastic Ensemble Value Expansion. In NeurIPS, 2018.

The most closely related work I’m aware of is the following:

Deits et al., Lvis: Learning from value function intervals for contact-aware robot controllers. In ICRA, 2019.

The authors should clarify their contributions with respect to these papers. More importantly, the authors should discuss their motivation for indirectly learning a Lyapunov function instead of simply learning the value function (which appears to be more natural and potentially more effective).

Next, the authors’ experiments are very weak. They only consider two environments, the inverted pendulum and car, both of which are very simple. The inverted pendulum starts near the unstable equilibrium, which further trivializes the problem. In addition, they do not even appear to give the dynamics model of the car they are using (or the state space).

Finally, this paper is poorly written and hard to follow. They provide a lot of definitions and equations without sufficient explanation or justification, and introduce a lot of terminology without giving sufficient background.

------------------------------------------------------------------------------------------------------------------------------------------------

Post rebuttal: While I appreciate the authors' comments, they do not fundamentally address my concerns that the paper is too unclear in terms of the meaning of its technical results to merit acceptance. As a concrete example, in their clarification, the authors indicate that they obtain "probabilistic safety guarantees" by checking the Lyapunov condition (5) using sampling. However, at best, sampling can ensure that the function is "approximately" Lypaunov (e.g., using PAC guarantees) -- i.e., satisfies (5) on all but 1-\epsilon of the state space.

Unfortunately, an "approximately" Lyapunov function (i.e., satisfies the Lyapunov condition (5) on 1-\epsilon of the state space) provides *zero* safety guarantees (not even probabilistic safety at any confidence level). Intuitively, at each step, the system has a 1-\epsilon chance of exiting a given level set of the Lyapunov function. These errors compound as time progresses; after time horizon T, only 1 - T * \epsilon of the state space is guaranteed to remain in the level set, so eventually the safety guarantee is entirely void.

One way to remedy this is if the Lyapunov function is Lipschitz continuous. However, then, the number of samples required would still be exponential in the dimension of the state space. At this point, existing formal methods tools for verifying Lyapunov functions would perform just as well if not better, e.g., see:

Soonho Kong, Sicun Gao, Wei Chen, and Edmund Clarke. dReach: δ-Reachability Analysis for Hybrid Systems. 2015.

This approach was recently applied to synthesizing NN Lyapunov functions (Chang et al. 2019). My point isn't that the authors' approach is invalid, but that given the current writing it is impossible for me to understand the theoretical properties of their approach.

Overall, I think the paper may have some interesting ideas, but I cannot support publishing it in its current state

---

> ### Author Response · Authors · 2020-11-17
> **Reviewer1 Response Part 1- Thank you for the feedback**
>
> We thank the reviewer for their feedback and hope that we address the concerns raised in our reply.
> Our work addresses the problem of designing a controller to maximize the stability of the closed-loop system. The objective in this work is to combine learning from data and control theory to train a controller that has provable safety certificates in terms of Lyapunov functions.
>
> Regarding clarity, the paper borrows terms and definitions from control theory which are instrumental to motivate the proposed algorithm. While it is not practical to write explanations to all the standard results in control systems and Lyapunov stability, we have tried our best to elucidate the relevant background in Section 2. For more information on Lyapunov-based control methods and certification, we would like to refer the reviewer to Bobiti (2017) and Limon (2003, 2009) which we have also cited in the paper. We are open to further suggestions by the reviewer on how to improve the readability of the work.
>
> Please find our comments on the specific questions/concerns that have been raised:
>
> #### 1. Regarding the problem statement
>
> Restating the approach rationale specified on page 2, our work aims to match or enlarge the safe region of an unknown controller. We aim to design a stabilizing controller using data collected (one-step demonstrations) from an unknown controller (demonstrator) such that the largest possible stability region is obtained. In an ideal case, the region obtained by the new controller should match or extend the one from the original unknown one (demonstrator). However, due to function approximation, this may not always be possible. In this work, we offer a framework to produce a verifiably safe new controller while at the same time attempting to match the size of the stable region of the demonstrator as much as possible.
>
> #### 2. Regarding learning of the Lyapunov function in the algorithm
>
> As specified in Section 3.1, we design the Lyapunov neural network (NN) such that it is Lipchitz and satisfies the Lyapunov conditions (4). This is described in more detail in the work from Richards et.al (2018).
>
> Provided that condition (5) is verified, the stability guarantees provided in Theorem 1 (formerly called Lemma 1) are that for an MPC which uses the Lyapunov NN function as its terminal cost. Proving that a NN trained via SGD is “exact” at inference time or bounding its test error during training are open areas of research. As stated on page 3, in our work, we use a-posteriori sampling-based verification of the neural Lyapunov function to verify (5) at each stage of our alternate learning algorithm. This provides a high probability certificate that the network is a Lyapunov function according to the conditions (4) and (5), as described in Bobiti (2017). Due to page constraints, a detailed description of the algorithm used for verification is specified in Appendix E.
>
> #### 3. Regarding robustness margins to model errors
>
> We thank the reviewer for bringing this to our notice. In the appendix, we have shown a bound for the maximum model error for which the closed-loop system can retain a given Input-to-State Stability (ISS), i.e. robustness to additive errors. Due to space limitations, we had not included this bound as part of the theorem in the main paper. However, we recognize that it would be beneficial to state this result and will do so in the updated version of the paper.
>
> The bound on the maximum model error for ISS tells us that, for a stronger contraction factor and a greater safe set size, we can proportionally tolerate more uncertainty on the model. However, the effect on the stability performance of the controller is highlighted in Theorem 2 (formerly Lemma 2), where we discuss how the model error affects the final radius of convergence of the cost (and implicitly the control error). The bound presented there shows that it is beneficial to decrease the horizon length if the uncertainty in the model is large and the system to be controlled has large Lipschitz constants. In our experiments, we use this result in designing the controller by reducing its horizon length to one and still showing that ensures stability.

---

> > ### Author Response · Authors · 2020-11-17
> > **Reviewer1 Response Part 2**
> >
> > #### 4. Regarding boundedness of model error
> >
> > We justify the boundedness assumption on the surrogate dynamics model by assuming local Lipschitz-ness and bounded domains. In practice, we need milder conditions (like local continuity) to have a bounded error. The existence of these margins is important to characterize the performance of a controller.
> >
> > In our algorithm, we account for the model error implicitly through the formulation of the MPC. As discussed in Theorem 1.6, the ISpS stability gain of the MPC (which is related to the performance bound of Theorem 2) increases monotonically with the model error and horizon length. In our experiments, we show that learning a Lyapunov function makes it possible to reduce the horizon length of the MPC to as low as one, thereby mitigating the accumulation of errors due to an imperfect dynamics model. We will clarify the relation between Theorems 1 and 2 in the updated version.
> >
> > #### 5. Regarding the loss functions used for training
> >
> > The intuition behind the loss function used for training the Lyapunov NN is justified more clearly in prior work by Richards et al. (2018). We would like to refer the reviewer to the paper as well as the comments provided to Reviewer 2 for more details. In the existing literature, we are not aware of any other loss functions that allow learning a safe Lyapunov level set. However, we are open to suggestions regarding the same from the reviewer.
> >
> > #### 6. Comparison between value function and Lyapunov function
> >
> > We disagree with the reviewer here. A ‘learned’ value function surrogate is not always an exact value function. Thus, it is not guaranteed to be a valid Lyapunov function unless one can prove the learning convergence, which is hardly possible using function approximation. In principle, one can learn a value function and apply formal verification to it; however, we believe incorporating the stability verification directly in the learning objective (when possible) is more direct. Learning a value function is not simple in general, and large errors could occur for simple biases, which are not essential in the context of the Lyapunov function, where all that matters is the decrease and not the value itself. This decrease is vital for safety and cannot be guaranteed by value learning using function approximation and finite computation, as far as the authors are aware.
> >
> > #### 7. Regarding experiments details
> >
> > Due to page constraints, we provided detailed descriptions of the environment models in Appendix C. However, we state in the main paper that the pendulum is torque-limited and can be controlled by starting only within $\pm60$ degrees from the upper position. This is achieved better by our controller in comparison to other baselines. The car may seem simple however its non-linearity and the constraints make it particularly challenging for the baselines to solve.
> >
> > References:
> > 1. Bobiti, Ruxandra Valentina. Sampling–driven stability domains computation and predictive control of constrained nonlinear systems. Diss. PhD thesis, 2017. URL: https://pure.tue.nl/ws/files/78458403/20171025_Bobiti.pdf
> > 2. Limon, D., T. Alamo, and E. F. Camacho. "Stable constrained MPC without terminal constraint." Proceedings of the 2003 American Control Conference, 2003.. Vol. 6. IEEE, 2003.
> > 3. Limon, D., et al. "Input-to-state stability: a unifying framework for robust model predictive control." Nonlinear model predictive control. Springer, Berlin, Heidelberg, 2009. 1-26.
> > 4. Richards, Spencer M., Felix Berkenkamp, and Andreas Krause. "The lyapunov neural network: Adaptive stability certification for safe learning of dynamical systems." arXiv preprint arXiv:1808.00924 (2018). URL: https://arxiv.org/abs/1808.00924

---

> > > ### Author Response · Authors · 2020-11-23
> > > **Future work and clarifications on value learning**
> > >
> > > We thank again the reviewer for their feedback. Following discussions with all the reviewers, we believe we could further highlight in  the paper that the proposed approach could also be extended in future work by complementing the Lyapunov loss with a more "classic RL" value function learning loss. The rationale behind this would be to balance off the different terms in the error bound of Theorem 2 and perhaps tradeoff the stability (contraction) property with infinite-horizon optimality. We think this is a very interesting avenue as well as choosing the best horizon length that minimises the suboptimality bound which was proposed by Reviewer 4. We will make further amendments as soon as possible to the paper to emphasize this. In the meantime, we hope the reviewer could reconsider their score. We remain available to further discuss and amend the paper accordingly.

---

### Official Review · AnonReviewer2 · 2020-11-02
**Paper presents solid control theoretical foundation, though unsure how novel the ideas are**

**Rating:** 5
**Confidence:** 4

**Review:**

In this paper the author proposed an MPC algorithm in which both the dynamics function and the Lyapunov function are parameterized with neural networks.. Specifically leveraging the results of Lyapunov networks (2018 CORL paper: https://arxiv.org/abs/1808.00924) for learning Lyapunov functions, the authors derived an MPC algorithm for quadratic cost/reward problems and also proved the stability, robustness, and sub-optimality performance. To demonstrate the effectiveness of the algorithms, the authors also evaluated this approach on the simple inverted pendulum and car kinematics tasks.

In general I find this paper presents a comprehensive results of a model-based control method that is very popular in the control theory community. To justify their algorithms they also proved several standard properties (stability, sub-optimality performance) in control, which I appreciate their efforts. However, I do have severals questions/concerns regarding the details of their approach:

1) The presentation of the loss function of Lyapunov network is not easy to parse, especially there are couple terms that contain specific mathematical operators (sign, ReLU). Can the authors explain each term in the loss and why such choices of loss terms are necessary. Is this loss function identical to the Lyapunov network 2018 CORL paper?

2) From the main paper it is unclear how the NN-dynamics model \hat f is learned. Does it just train based on prediction loss? More importantly, while the MPC algorithm uses the learned model how does the dynamics model error affect the stability/robustness/performance bounds of the control algorithm? I cannot immediately find this information in lemma 1 and lemma 2, which makes me worried about the correctness of these results. (Unfortunately I haven't had a chance to check the appendix for proofs)

3) Having sub-optimality performance for MPC algorithms is a nice result, as not many MPC algorithms have performance guarantees. However these kind of results are also not new (for example, see https://ieeexplore.ieee.org/document/4639448).  How does the MPC performance result here compared with the ones by Grune and Rantzer?

4) Among various safe MPC papers, how does the proposed one in this paper compared with this safe MPC algorithm: https://arxiv.org/pdf/1803.08287.pdf, which is also proposed by Andreas Krause's group (that proposed the Lyapunov network)? At least experimentally how does the proposed algorithm compare with other safe MPC baselines (such as the one above) on the standard benchmark tasks (for example the above work also tested the algorithms on the pendulum task).

On the overall, I find this paper's algorithm interesting. However, there are several technical question listed above, and one high-level concern is its novelty. Without further discussions, it appears to me that the work combines several existing results on Lyapunov network and MPC, for which the contribution is rather incremental.

---

> ### Author Response · Authors · 2020-11-17
> **Reviewer2 Response Part 1- Thank you for the feedback**
>
> We thank the reviewer for their constructive feedback and are pleased to hear that our proposed algorithm was found interesting. We address the raised points in the following reply and will make the necessary changes in the uploaded version as well. We hope these answer the remarks by the reviewer and convince them to improve the score. We are of course open to further iterations and amendments to the paper.
>
> Please find our comments on the specific questions/concerns that have been raised:
>
> #### 1. Regarding the novelty of the work
>
> As mentioned in the paper, we extend the theoretical results from Limon et al. (2003) to the discounted setting and the use of  lambda-constractive Lyapunov functions; those from Lowrey et al. (2018) are extended to the uncertain model setup. The theorems presented in our paper aim to motivate the proposed algorithm which we believe has never been presented in the literature.
>
> #### 2. Regarding loss function for learning Lyapunov NN:
>
> The loss proposed in our paper is similar to the one by Richards et al. (2018). The first part encourages the function to decrease over trajectories from within the estimated safe set. The second part is aimed at estimating the function level that defines the safe set. The main difference in this loss function from that by Richards et al. (2018) is in this second part. Instead of using stability labels obtained by performing a forward propagation of T steps and verifying convergence, our algorithm only performs a unitary time-step forward propagation. This is mentioned at the beginning of page 6 in Section 3.3, however, we will reinforce the statement in the new upload. We use the Lyapunov network itself to generate pseudo-labels (stability certificate via $sign(\Delta V)$). This is more data-efficient as we don’t need to have all trajectories reach completion, and is less prone to error accumulation than using a long-horizon simulation with an imperfect surrogate model (Richards et al. (2018) use the true dynamics model for forward propagation).
>
> Besides that, in the training procedure, we verify the Lyapunov network on a validation dataset and perform cross-validation based on the number of verified and not verified points. We also perform a posteriori formal verification (the algorithm for this is presented in Appendix E).  We will make these points clearer in the updated version.
>
> Notation-wise: $ReLU(x)=max(x,0)$ and $sign(\cdot)$ is the sign of the quantity which returns either 1 or -1. We will clarify this in the paper as well.
>
> #### 3. Regarding surrogate dynamics model:
>
> We do not require the surrogate dynamics model to be trained in a particular way.  Further, it is not essential that the model is parametrized by a neural network, as long as its function class has Lipschitz continuity. Since all the sets are bounded, the model’s error bound can be inferred based on the Lipschitz constants (as discussed on page 3). However, in our algorithm, we assume that the learned dynamics model is given and its error bound is known.
>
> We will add a worst-case one-step error bound for ISS of the controller in the revised version of Theorem 1 (formerly Lemma 1). This margin was previously stated in the proof of the theorem (in Appendix A), however, we will move it to the main paper.
> Due to page constraints, we specified the details about the system models and the training of the dynamics model in Appendix D, while providing the information about the setup in the main paper (Section 4). The NN-dynamics surrogate model $\hat{f}$ is trained using transition tuples (this is stated above eq. (8) on page 3). The data to train the model is collected using a random policy as typically done in system identification. This is mentioned on pages 6-7 for both experiments.
>
> #### 4. Regarding related work on MPC suboptimality:
>
> We thank the reviewer for pointing us to this important paper.  We cited a different work from these authors on the stability of optimal control with discount factors (Gaitsgory et al., 2015) which is closely related to our discounted setup. The work by Gruene and Rantzer (2008), however, is indeed also related and offers a key set of results for MPC suboptimality under the assumption that the loss satisfies an exponential controllability condition. In our paper, we consider a discounted setting which is more common in RL than in controls (because of stability limitations as we highlight in Theorem 1). Our results are built upon those from Lowrey et al.( 2018).  However, we will add the missing reference (Gruene and Rantzer, 2008) to our related work section in the revised upload.

---

> > ### Author Response · Authors · 2020-11-17
> > **Reviewer2 Response Part 2**
> >
> > #### 5. Comparison to work by Koller et al. (2018)
> >
> > Koller et al. (2018) rely on Gaussian Process (GP) and RHS kernels to obtain conservative closed-form results. GPs typically do well in the low-data regime and are generally limited to low-dimensionality.
> > While we consider the work by Koller et al. (2018) as an important contribution (and will add to the related works), we believe comparing our approach to it is not straightforward since the two look at orthogonal problems. In our work, we train the dynamics model (a NN) and Lyapunov network offline on a larger dataset. We don’t perform exploration or online learning. On the other hand, Koller et al. (2018) focus more on safe exploration with very few data points collected online which are used to update a GP model. For a comparison of NNs and GPs in the context of online learning, we instead refer the reviewer to Gal et al. (2016).
> >
> > References:
> > 1. Gaitsgory, Vladimir, Lars Grüne, and Neil Thatcher. "Stabilization with discounted optimal control." Systems & Control Letters 82 (2015): 91-98.
> > 2. Gal, Yarin, Rowan McAllister, and Carl Edward Rasmussen. "Improving PILCO with Bayesian neural network dynamics models." Data-Efficient Machine Learning workshop, ICML. Vol. 4. 2016. URL: http://mlg.eng.cam.ac.uk/yarin/PDFs/DeepPILCO.pdf
> > 3. Grune, Lars, and Anders Rantzer. "On the infinite horizon performance of receding horizon controllers." IEEE Transactions on Automatic Control 53.9 (2008): 2100-2111.
> > 4. Koller, Torsten, et al. "Learning-based model predictive control for safe exploration." 2018 IEEE Conference on Decision and Control (CDC). IEEE, 2018.
> > 5. Limon, D., T. Alamo, and E. F. Camacho. "Stable constrained MPC without terminal constraint." Proceedings of the 2003 American Control Conference, 2003.. Vol. 6. IEEE, 2003.
> > 6. Lowrey, Kendall, et al. "Plan online, learn offline: Efficient learning and exploration via model-based control." arXiv preprint arXiv:1811.01848 (2018).
> > 7. Richards, Spencer M., Felix Berkenkamp, and Andreas Krause. "The lyapunov neural network: Adaptive stability certification for safe learning of dynamical systems." arXiv preprint arXiv:1808.00924 (2018). URL: https://arxiv.org/abs/1808.00924

---

> > > ### Author Response · Authors · 2020-11-23
> > > **Further clarification on scope and novelty**
> > >
> > > We thank again the reviewer for their feedback and have revised the paper accordingly.
> > >
> > > We would like to further reinforce the case for the paper on novelty, which we believe goes beyond the extension of the theorems from the related work (also a contribution) but lies in the algorithmic development: the use of alternate learning with an 'epsilon extended' target ROA, the (required) cross-validation and formal verification to extend the ROA of a learned controller from an unknown demonstrator, despite the (unknown but limited) model error and the use (first in literature) of a NN Lyapunov function as the terminal cost for MPC. Lastly, the use of one step unlabelled transitions instead of long labelled sequences for learning the terminal cost.
> > >
> > > We hope the reviewer could still reconsider their score. We are available for further interaction and amendments to the paper if required by the reviewer.

---

### Decision · Program_Chairs · 2021-01-07
**Final Decision**

**Decision:**

Reject

**Comment:**

The authors propose an MPC based approach for learning to control systems with continuous state and actions - the dynamics, control policy and a Lyapunov function are parameterized as neural networks and the authors claim to derive stability certificates based on the Lyapunov function.

The reviewers raised several serious technical issues with the paper as well as the lack of clarity in the presentation of the main technique in the initial version of the paper. While the clarity concerns were partially addressed during the rebuttal, the technical concerns (in particular those raised by reviewer 1) remain unaddressed - the stability certificate derived is questionable due to the fact that sampling based approaches to certifying that a function is a valid Lyapunov function are insufficient to derive any stability guarantee. Further, the experimental results are only demonstrated on relatively simple dynamical systems. Hence I recommend rejection.

However, all reviewers agree that the ideas presented in the paper are potentially interesting - I would suggest that the authors consider revising the paper to address the feedback on technical issues and submit to a future venue.